# CAN WEAK QUANTIZATION MAKE WORLD MODELS PHYSICALLY INTERPRETABLE?

## ABSTRACT

Deep learning models are increasingly employed for perception, prediction, and control in autonomous systems. For achieving realistic and consistent outputs, it is crucial to embed physical knowledge into their learned representations. However, doing so is difficult due to high-dimensional observation data, such as images, particularly under conditions of incomplete system knowledge and imprecise state sensing. To address this, we propose *Physically Interpretable World Models*, a novel architecture that aligns learned latent representations with real-world physical quantities. To this end, our architecture combines a physical interpretable image autoencoding model and a partially known learnable dynamical model. We conduct an in-depth analysis of the latent space, evaluating the effects of continuous versus discrete representations, as well as intrinsic versus extrinsic physical interpretable encodings. The training incorporates weak distributional supervision to eliminate the impractical reliance on ground-truth physical knowledge. Through three case studies, we demonstrate that our approach not only provides physical interpretability but also achieves state prediction accuracy superior to state-of-the-art models, thus advancing interpretable representation learning.

## 1 INTRODUCTION

Accurate and robust trajectory prediction from high-dimensional sensor data, such as camera images, is a fundamental challenge for the safe operation of autonomous systems. A dominant paradigm for this task is to learn a compact latent representation of the environment and evolve it over time. This principle forms the basis of modern *world models* (Ha & Schmidhuber, 2018), which extend Variational Autoencoders (VAEs) (Kingma, 2013) by integrating predictive components like recurrent neural networks (RNNs) to capture temporal dependencies (Mao et al., 2024b; 2025b). Recent advancements in hierarchical latent modeling and transformer architectures have further enhanced the fidelity and long-horizon capabilities of these predictive models (Micheli et al., 2022; Hafner et al., 2023; Seo et al., 2023). However, while these models achieve impressive predictive performance, their learned representations often function as a "black box," lacking a clear connection to the underlying physical state of the system.

Bridging this gap would greatly improve the trustworthiness and controllability of autonomous systems operating in complex, high-risk scenarios. For instance, in autonomous driving, linking latent representations to physical states could generate causal explanations for decisions (e.g., slowing down due to occlusions) and enable high-assurance methods like formal verification (Hasan & Tahar, 2015) and run-time shielding (Waga et al., 2022). Moreover, physical states are essential for high-assurance methods such as formal verification (Katz et al., 2022) and run-time shielding (Alshiekh et al., 2018). For instance, if a predicted position lies in an occupied lane, a shield can intervene to prevent a collision. Physically interpretable representations also support causal explanations. For example, "that car won't slow down because it doesn't see you" which enhances trust in autonomous behavior. Finally, physical priors improve generalization and sample efficiency by guiding the model toward plausible trajectories.

So far, several approaches for achieving physically interpretable latent spaces from high-dimensional observations can be categorized into two fundamental paradigms: intrinsic and extrinsic encoding methods (Figure 1a). Extrinsic approaches adopt a two-stage strategy, first learning abstract latent representations from images through standard autoencoders, then mapping these representations to

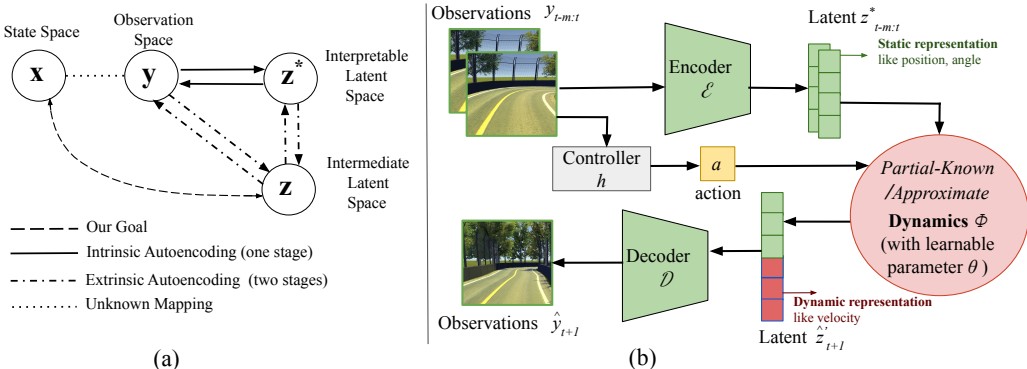

(a)            (b)

Figure 1: Overview of Physically Interpretable World Models (PIWM). (a) Two approaches to learn a physically meaningful latent space ($z^*$): Intrinsic and Extrinsic Autoencoding. (b) The PIWM architecture learns a structured latent representation $z^*$ from images, uses a learnable dynamics model $\phi$ with physical priors to predict future latent states, and decodes them into future images.

physical quantities via additional neural network layers. While this decoupling provides flexibility, it typically requires ground-truth physical states for training the mapping function and may lose physical consistency during the intermediate representation stage. Intrinsic approaches attempt to directly encode physical structure into the image encoder itself, ensuring that the latent representations extracted from visual inputs inherently correspond to meaningful physical quantities (Le et al., 2025). However, both extrinsic and intrinsic methods face significant limitations: they require large-scale datasets with precise physical annotations, need exact supervision of physical variables during training, or are limited to scenarios where object decomposition is feasible and meaningful with object-centric representation learning method (Mosbach et al., 2025).

We aim to address the problem of learning physically meaningful representations *without exact supervision of physical labels* in the context of trajectory prediction. Instead, we rely on weak supervisory signals, which contain valuable information such as approximate speed bounds inferred from coarse or noisy position sequences. As a natural representation of such uncertain signals, we use *distributions* over the true values, which tend to come from sensing and perception pipelines in autonomous systems. In practice, many such pipelines (e.g., GPS, radar) produce probabilistic estimates, such as confidence ellipses or ranges, rather than precise values. Distributions offer a simple, expressive, and widely used abstraction for representing such uncertainty, especially in field robotics, where full state information is rarely available. Thus, we use weak-supervision distributions to steer the encoding of high-dimensional images into physically interpretable representations. This creates an opportunity for physics-informed prediction. Since the governing dynamics of many real-world systems are known or can be well-approximated, we can now make physically plausible predictions by embedding this known structure and learning only its unknown internal parameters.

Our central contribution is the *Physically Interpretable World Model* (PIWM), a novel and flexible architecture designed to learn physically meaningful representations from high-dimensional observations. The core principle of PIWM is to align its learned latent space with real-world physical quantities. It achieves this by integrating a learnable dynamics model, which incorporates partially known physical equations as structural priors, with a powerful image autoencoder. A key aspect of our approach is the use of weak distributional supervision to guide this alignment, which eliminates the impractical reliance on ground-truth physical state annotations.

Within the PIWM architecture, we conduct an in-depth experimental analysis to answer a central question: Can weak quantization make world models physically interpretable, and how to construct a more interpretable latent space? To this end, we systematically evaluate several key design choices, including continuous versus discrete latent spaces and the use of intrinsic versus extrinsic physical encodings. We validate the proposed architecture across three autonomy case studies: cart pole, lunar lander, and autonomous racing donkey car. The experiments demonstrate that the PIWM framework not only achieves superior physical grounding and lower prediction error compared to purely data-driven baselines but also provides clear insights into the optimal design of physically interpretable latent spaces. More specifically, our results reveal that extrinsic architectures, which decouple visual perception from physical interpretation, significantly outperform their end-to-end intrinsic counterparts in both predictive accuracy and the plausibility of the learned physical parameters.

## 2 PRELIMINARIES

### 2.1 AUTONOMOUS SYSTEM AND WORLD MODELS

**Definition 1** (Autonomous System). *A autonomous system $s = (X, I, A, \phi_\theta, g, Y)$ models the evolution of an image-based control system, where the state set $X$ defines the finite-dimensional space of physical states, the initial set $I \subset X$ specifies possible starting states, and the action set $A$ contains all possible control actions. The system dynamics $\phi_\theta : X \times A \times \Theta \to X$ govern state transitions under physical parameters $\theta \in \Theta$, the observation function $g : X \to Y$ maps states to observations, and the controller $h : Y \to A$ selects actions based on observations.*

Consider an autonomous system with observation-based control, the states of which evolve as $x_{t+1} = \phi(x_t, a_t, \theta^*)$, where $\theta^*$ are the true (unknown) physical parameters. The known controller $h$ relies on observations $y$ from sensors (e.g., cameras). Such controllers can be trained by imitation learning (Hussein et al., 2017) or reinforcement learning (Kaelbling et al., 1996; Lillicrap, 2015). Given an initial state $x_0 \in I$, a *state trajectory* is defined by iteratively applying the dynamics function $\phi$, yielding a sequence $(x_0, x_1, \ldots, x_{i+1})$, where each state is $x_{t+1} = \phi(x_t, a_t, \theta^*)$ and the action is generated by the controller based on the observation: $a_t = h(y_t) = h(g(x_t))$. Correspondingly, an *observation trajectory* $(y_0, y_1, \ldots, y_{t+1})$ is generated by applying the observation function $g$ to each state: $y_t = g(x_t)$.

We consider the setting where the state is not directly observable, and to anticipate hazards and adapt, the system needs to forecast observations. This is done by combining a world model conditioned on past observations with an observation-based controller $h$, essentially forming an efficient online simulator of future observations.

**Definition 2** (World Model). *A world model $\mathcal{W} = (\mathcal{E}, f, \mathcal{D})$ predicts the future evolution of observations in a robotic system $s$, where the encoder $\mathcal{E} : Y \to Z$ compresses high-dimensional observations into latent representations, the predictor $f : Z \times A \to Z$ forecasts future latents conditioned on actions, and the decoder $\mathcal{D} : Z \to Y$ reconstructs predicted observations. The actions $a_t$ are generated by a controller $h : Y \to A$ based on the current observation $y_t$, and together with the encoded latent $z_t = \mathcal{E}(y_t)$, are used to obtain the future latent $z_{t+1} = f(z_t, a_t)$. The predicted observation $\hat{y}_{t+1}$ is then obtained by decoding $z_{t+1}$ with $\mathcal{D} : \hat{y}_{t+1} = \mathcal{D}(z_{t+1})$.*

While world models enable diverse tasks (e.g., controller training), they are usually evaluated with *predictive accuracy* — how closely the predicted observations $\hat{y}_{t+1:t+n}$ match the true observations $y_{t+1:t+n}$. The typical metrics include mean squared error (MSE) and structural similarity index (SSIM). Unfortunately, these metrics are inherently limited: similar pixel-wise observations may correspond to very different underlying states. This means that the similarity (low MSE and high SSIM) between $\hat{y}_{t+1:t+n}$ and $y_{t+1:t+n}$ does not guarantee similarity between the underlying physical states corresponding to the predicted observations and the true physical ones. Thus, the error in predicted observations is not physically interpretable. This severely limits the predictions' utility for downstream tasks like safety monitoring and planning.

### 2.2 INTERPRETABLE REPRESENTATION LEARNING AND PREDICTION

The autoencoder components $\mathcal{E}$ and $\mathcal{D}$ in world models are primarily based on either VAE, VQ-VAE, or their variants. VAEs (Kingma, 2013) encode high-dimensional inputs $y$ into continuous latent vectors $z = \mathcal{E}(y)$, which are then decoded back into reconstructed inputs $\hat{y} = \mathcal{D}(z)$. A standard VAE assumes a simple prior distribution over the latent space, typically an isotropic Gaussian, which facilitates tractable inference but imposes a strong inductive bias. A VQ-VAE (Van Den Oord et al., 2017) discretizes the latent space by mapping a high-dimensional observation $y_t$ to a continuous latent vector $z_t = \mathcal{E}(y_t)$, then quantizing it via a *codebook* $\{\mathbf{e}_k\}_{k=1}^K$ to obtain a discrete latent *index* $z_t^* = \arg\min_k \|z_t - \mathbf{e}_k\|_2^2$ closest to the continuous latent $z_t$. The decoder then reconstructs the observation via $\hat{y}_t = \mathcal{D}(\mathbf{e}_{z_t^*})$. While discretization introduces structure and regularization, the learned codebook vectors $\mathbf{e}_k$ themselves are typically unstructured and lack interpretation. Therefore, merely discretizing the space is insufficient; hence, an explicit mechanism is needed to align these latent representations with physical semantics.

Our setting assumes that general knowledge about the dynamics function $\phi$, namely the structure of its equations. However, the dynamics parameters $\theta$, such as the mass or friction coefficient, are

unknown, making the system $s$ only partially specified. In practice, it is often hard to find the true parameters $\theta^*$ because true physical states $x$ cannot be measured precisely. However, it is typical to compute a distribution $p(x)$ from high-dimensional observations. For instance, the robot's pose may be estimated from images as a range/distribution — and serve as a weak supervisory signal.

Thus, to train a world model, we are given a dataset $\mathcal{S}$ of trajectories, consisting of $N$ sequences of length $M$, where each sequence contains tuples of images, actions, and state distribution labels:

$$\mathcal{S} = \left\{ \big( y_{1:M}, a_{1:M}, p_{1:M}(x) \big) \right\}_{1:N} \tag{1}$$

The weak supervision is provided by state distributions $p_t(x)$, for which we assume no analytical form is known. Instead, we can only draw a finite set of samples from them. This represents a challenging yet practical scenario, as it is a weaker assumption than knowing the distribution's type (e.g., Gaussian), its parameters (e.g., mean and variance), or even a definitive interval.

Our ultimate task is to learn a representation $z^*$ that accurately approximates the system's true, unobserved physical state $x$ for prediction. We formalize this as two distinct but related problems:

**Definition 3** (Interpretable Representation Learning Problem). *Given a dataset $\mathcal{S}$ and a controller $h$, the objective is to* learn a state representation $z^* = \mathcal{E}(y)$ that minimizes the mean squared error *to the true physical state $x$:*

$$\min_{\mathcal{E}} \; \mathbb{E}\big[\|x - \mathcal{E}(y)\|_2^2\big] \tag{2}$$

The prediction problem is to forecast the evolution of these interpretable representations over time.

**Definition 4** (Prediction Problem for Interpretable Representations). *Given a dataset $\mathcal{S}$, a dynamics function $\phi$ with unknown parameters, and a controller $h$, the objective is to* train a predictor *that maps a history of representations to a future representation, $\hat{z}^*_{t+k} = \mathcal{P}(z^*_{t-m:t})$, by minimizing the future state prediction error:*

$$\min_{\mathcal{P}} \; \mathbb{E}\big[\|x_{t+k} - \hat{z}^*_{t+k}\|_2^2\big]. \tag{3}$$

The objectives in Definitions 3 and 4 are formulated with respect to the true physical state $x$, which serves as the ultimate ground truth for evaluating physical interpretability. However, since $x$ is not accessible during training, these objectives cannot be optimized directly. Our proposed method, detailed in Section 3, addresses this challenge by constructing a tractable surrogate objective that leverages the weak supervision by sampling state distributions $p(x)$.

## 3 ARCHITECTURE

We introduce the *Physically Interpretable World Model* (PIWM), a flexible prediction architecture with physically-grounded representations of high-dimensional observations. It consists of two core components: (1) a physically interpretable autoencoder responsible for learning the state representation, and (2) a learnable dynamics model that predicts the evolution of this representation.

**Learning Physically Interpretable Representations.** The primary goal of the autoencoder is to map a high-dimensional observation $y$ to a low-dimensional interpretable latent state $z^*$. This representation must be explicitly aligned with the true physical state $x$. Per Section 2, we cannot access $x$ or the analytical form of its supervisory distribution $p(x)$. Instead, we are given access to a set of $L$ state proxy samples, $\Xi = \{\xi^{(l)}\}_{l=1}^L$, drawn from $p(x)$. To leverage these proxy samples $\Xi$ for training, we formulate a general interpretability loss, $\mathcal{L}_{\text{interp}}$, which measures the discrepancy between a predicted physically interpretable state $z^*_p$ and the sample set $\Xi$. In our experiments, we primarily use a Mean Squared Error (MSE) formulation that penalizes the distance to the empirical mean of the samples:

$$\mathcal{L}_{\text{interp}}(z^*_p, \Xi) = \|z^*_p - \hat{\mu}_\xi\|_2^2, \quad \text{where} \quad \hat{\mu}_\xi = \frac{1}{L}\sum_{l=1}^L \xi^{(l)}. \tag{4}$$

An alternative could be to use the Kullback-Leibler (KL) Divergence against a Gaussian fitted to the samples. This serves as the mechanism for enforcing physical grounding throughout our models.

A central design question in learning representation $z^*$ is how to manage two competing objectives: reconstructing high-dimensional observations versus aligning the latent space with low-dimensional

physical states. We consider two approaches. The *Intrinsic* approach attempts to achieve both objectives simultaneously with a single, end-to-end encoder. While potentially more efficient, this forces one network to both capture fine-grained visual details (for reconstruction) and ignore those same details to extract the underlying physical state — a difficult optimization task. In contrast, the *Extrinsic* approach follows a two-stage process: a vision autoencoder first learns an intermediate representation focused on reconstruction, and then a second, physical encoder extracts the interpretable state from it. This modularity may stabilize training but risks information loss in the intermediate step. Given the fundamental trade-offs, we will systematically investigate both approaches.

Orthogonal to this architectural choice, a second key design decision is the nature of the latent space $Z^*$. *Continuous spaces* can represent high-fidelity physical quantities but lack a built-in organizational prior, requiring explicit regularization for disentanglement. Conversely, *discrete spaces* enforce regularity by design through their finite codebook but lose precision due to quantization error. Below, we detail the combinations of intrinsic/extrinsic and continuous/discrete approaches.

**Intrinsic Autoencoding.** This approach employs a single, end-to-end encoder $\mathcal{E} : Y \to Z^*$ that directly maps an observation $y$ to the final interpretable latent state $z^*$. This unified representation must disentangle visual features from physical semantics, a known challenge where information may leak between different physical attributes and hinder the desired learning (Peper et al., 2025). For the case of *continuous* latent space, the encoder outputs the parameters for a posterior distribution $q(z^* \mid y)$ over the latent space $Z^*$. Sampled from this distribution, a latent vector is then partitioned in a fixed manner: $z^* = [z_p^*, z_v^*]$, where $z_p^*$ is the physically interpretable part and $z_v^*$ captures the remaining visual information necessary for reconstruction. The full objective follows the $\beta$-VAE formulation, with the interpretability loss applied only to $z_p^*$ and KL loss applied to $z_v^*$:

$$\mathcal{L}_{\text{intrinsic-cont}} = \mathcal{L}_{\text{recon}}(y, \hat{y}) + \lambda_{\text{interp}}\mathcal{L}_{\text{interp}}(z_p^*, \Xi) + \beta D_{\text{KL}}\big(q(z_v^*|y) \,\|\, \mathcal{N}(0, I)\big) \qquad (5)$$

For the case of *discrete* latent space, we structure the codebook to be interpretable. Each vector $\mathbf{e}_k$ is partitioned as $\mathbf{e}_k = [\mathbf{e}_k^p, \mathbf{e}_k^v]$. Only the visual part $\mathbf{e}_k^v$ is a typical learnable VQ-VAE codebook vector. The physical part $\mathbf{e}_k^p$ is a constant vector representing a specific point in a discretized grid of physical values (e.g., specific positions). Then, the interpretable state is computed as the average of the physical portions of the codebook vectors, $z_p^* = \frac{1}{|I|} \sum_{i \in I} \mathbf{e}_i^p$. The full objective combines the standard VQ loss with our interpretability loss:

$$\mathcal{L}_{\text{intrinsic-disc}} = \mathcal{L}_{\text{VQ}}(y, \hat{y}) + \lambda_{\text{interp}}\mathcal{L}_{\text{interp}}(z_p^*, \Xi), \qquad (6)$$

where $\mathcal{L}_{\text{VQ}}$ includes reconstruction, codebook, and commitment losses:

$$\mathcal{L}_{\text{VQ}} = \|y - \hat{y}\|_2^2 + \big\|\text{sg}[z_{\text{cont}}] - z_q\big\|_2^2 + \beta\big\|z_{\text{cont}} - \text{sg}[z_q]\big\|_2^2 \qquad (7)$$

Here, $z_{\text{cont}}$ is the continuous output of the encoder $\mathcal{E}$, and $z_q$ is its nearest vector from the codebook. The $\text{sg}[\cdot]$ is the stop-gradient operator, which ensures that gradients are routed correctly for the codebook loss (updating $z_q$) and the commitment loss (updating $z_{\text{cont}}$). The hyperparameter $\beta$ weights this commitment loss, controlling how strongly the encoder's output is encouraged to match the chosen codebook vector.

**Extrinsic Autoencoding.** This approach utilizes a two-stage training process to decouple perception from interpretation. First, a general-purpose vision autoencoder $(\mathcal{E}_v, \mathcal{D}_v)$ is trained to map an observation $y$ to an intermediate latent vector $z = \mathcal{E}_v(y)$. This stage is trained with a standard objective, independent of physical supervision. For the *continuous* case, this is a $\beta$-VAE trained to minimize:

$$\mathcal{L}_{\text{vision-cont}} = \mathcal{L}_{\text{recon}}(y, \hat{y}) + \beta D_{\text{KL}}\big(q(z \mid y) \,\|\, \mathcal{N}(0, I)\big) \qquad (8)$$

For the *discrete* case, a VQ-VAE is trained to minimize the standard VQ loss $\mathcal{L}_{\text{vision-disc}} = \mathcal{L}_{\text{VQ}}(y, \hat{y})$.

After the first stage, the vision encoder $\mathcal{E}_v$ is frozen. The second stage trains a separate, auxiliary physical autoencoder $(\mathcal{E}_p, \mathcal{D}_p)$ to map the intermediate representation $z = \mathcal{E}_v(y)$ to a final, purely physical representation $z^* = \mathcal{E}_p(z)$. The training objective for this stage is:

$$\mathcal{L}_{\text{physical}} = \lambda_{\text{interp}}\mathcal{L}_{\text{interp}}(z^*, \Xi) + \lambda_{\text{latent}}\mathcal{L}_{\text{recon}}\big(z, \mathcal{D}_p(z^*)\big) \qquad (9)$$

This same loss $\mathcal{L}_{\text{physical}}$ is used for both continuous and discrete intermediate representation $z$.

**Learnable Dynamics Model.** The second core component of our PIWM is a latent dynamics model $\phi$ that predicts the temporal evolution of the physically interpretable state $z^*$. Rather than using

black-box sequence models, our prediction is based on known dynamics equations, $\phi(z_t^*, a_t, \theta)$, where the form of $\phi$ (e.g., kinematics) is fixed and only its parameters $\theta$ are learnable. This allows our model to reflect the underlying physical laws while adapting to unknown system parameters $\theta$ such as friction and mass.

The dynamics model is responsible for estimating physical quantities that depend on a history of states, such as velocity, in order to predict the physics's evolution. To do this, the model is initialized with a short window of consecutive representations, $(z_t^*, z_{t+1}^*)$, produced by the encoder from observations. These two states, along with the control action $a_{t+1}$ that causes the transition from state $t + 1$ to $t + 2$, are used by the learnable dynamics model $\phi$ to predict the next state: $\hat{z}_{t+2}^* = \phi(z_t^*, z_{t+1}^*, a_{t+1}, \theta)$. By taking two consecutive states as input, the model can internally compute velocity and other time-derivative quantities necessary for an accurate physical prediction. After this initialization phase, the model can operate recursively for multi-step rollouts, taking its own prediction from the previous step as input to generate a future trajectory. This enables efficient, long-horizon forecasting without needing a sequence of observations at every step.

The learnable parameters $\theta$ of the dynamics model $\phi$ are trained by minimizing a dynamics loss, $\mathcal{L}_{\text{dyn}}$. This objective ensures that the predicted state $\hat{z}_{t+k}^*$, generated by recursively applying the dynamics model $\phi(\cdot, \cdot, \theta)$, aligns with the weak supervision available for that future time step. We use a Mean Squared Error (MSE) loss, which compares the prediction against the empirical mean of the proxy labels $\Xi_{t+k}$. The loss is defined as a function of the parameters $\theta$:

$$\mathcal{L}_{\text{dyn}}(\theta) = \|\hat{z}_{t+k}^* - \hat{\mu}_{\xi_{t+k}}\|_2^2, \tag{10}$$

where $\hat{z}_{t+k}^*$ is the state predicted by the dynamics model parameterized by $\theta$, and $\hat{\mu}_{\xi_{t+k}} = \frac{1}{L}\sum_{l=1}^{L} \xi_{t+k}^{(l)}$ is the empirical mean of the $L$ proxy samples for the future state. The full training algorithm, including gradient backpropagation through the dynamics, can be found in the Appendix.

## 4 EXPERIMENTAL EVALUATION

To validate our PIWM approach, we experiment on three environments: CartPole, Lunar Lander, and the DonkeyCar autonomous racing platform (Brockman et al., 2016; Viitala et al., 2021). These environments differ in the observation dimensionality, action space, and underlying dynamics.

**Experimental Setup.** For each environment, we collect a dataset of 60,000 trajectories, each with at least 50 time steps. To ensure diverse state space coverage, trajectories are generated by executing both random actions and those generated by well-trained neural controllers. The weak supervision signals are generated by perturbing the ground-truth physical states for different noise levels $\delta \in \{0, 5\%, 10\%\}$. For each, the weak supervision given by a uniform distribution over an interval constructed as follows. Its width is equal to the $\delta$ fraction of the full range of the respective state dimension. The interval's center is randomly shifted from the ground truth by an amount drawn from a uniform distribution over $[-\delta/2, \delta/2]$ of the full state range.

We evaluate our PIWM variants (Intrinsic/Extrinsic, Continuous/Discrete) against a suite of strong baselines under 5-fold cross-validation. We first include data-driven sequence models, an LSTM and a Transformer, which serve as non-physical benchmarks. Our primary comparisons are to state-of-the-art models in two categories. For the intrinsic approach, we compare against Vid2Para (Asenov et al., 2019) and GokuNet (Linial et al., 2021). For the extrinsic approach, we evaluate against DVBF (Karl et al., 2016). To specifically isolate and compare the performance of the latent dynamics predictors, we also include SindyC (Brunton et al., 2016) — a classic state-based method for dynamics discovery. For a fair comparison, both the DVBF and SindyC dynamics models operate on the interpretable latent representations produced by our continuous autoencoder. All models are configured to have a comparable parameter count to focus the comparison on architectural efficacy rather than model capacity. Further details and the results of CartPole can be found in the Appendix.

**Predictive Performance.** We first evaluate the primary task of long-horizon trajectory prediction. Figure 2 shows the root mean square error (RMSE) for 30-step future state prediction in the challenging Donkey Car environment, comparing both extrinsic and intrinsic methods across different levels of supervision noise, $\delta$. The results for extrinsic methods (Fig. 2) show a clear performance hierarchy. Our PIWM variants consistently outperform all baselines. The quantized extrinsic model (purple line) achieves the lowest and most stable prediction error, maintaining accuracy even as the

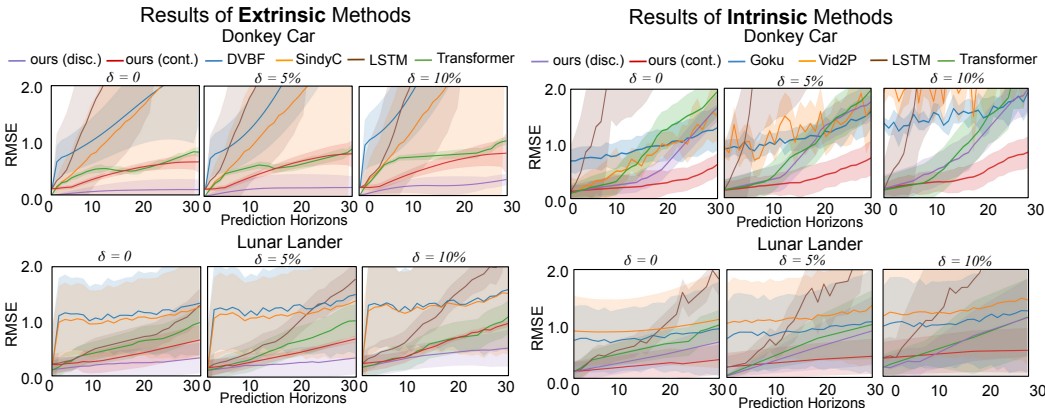

Figure 2: Prediction performance. The Root Mean Square Error (RMSE) of our PIWM variants (extrinsic methods, left; intrinsic methods, right) is compared against baselines over a 30-step prediction horizon in the Donkey Car and Lunar Lander across varying levels of weak supervision ($\delta$).

prediction horizon extends. Our continuous extrinsic model (red line) is the second-best performer. Both significantly surpass the extrinsic baselines (DVBF, SindyC) and the purely data-driven models (LSTM, Transformer), whose errors escalate rapidly.

A more nuanced picture emerges for the intrinsic methods (Fig. 2, right). Our PIWM models again demonstrate a clear advantage over the Vid2Para and GokuNet baselines. Strikingly, our continuous intrinsic model (red line) achieves a predictive accuracy that is highly competitive with, and in some cases even surpasses, our top-performing extrinsic models. This suggests that a well-regularized, end-to-end continuous architecture can be highly effective. In contrast, the quantized intrinsic model (purple line) exhibits less stability and higher error in this configuration, indicating that the optimization challenge of aligning a discrete codebook within a single, unified encoder is considerable. Nevertheless, both of our intrinsic variants outperform the baseline models, confirming the overall benefit of our training methodology. A key insight from these results is that decoupling visual perception from physical state inference (the extrinsic approach) is a critical design choice for achieving robust, long-term prediction. Furthermore, across both architectures, the quantized (discrete) latent space provides a powerful regularization effect, leading to more stable predictions than the continuous alternative, especially under noisy supervision.

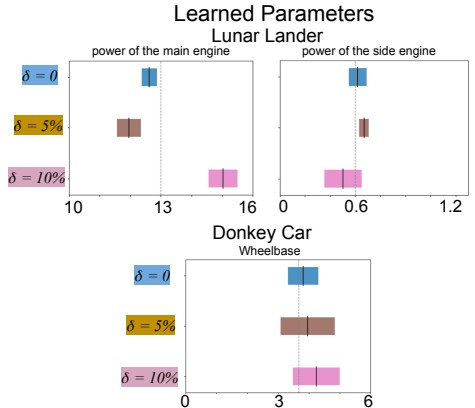

Figure 3: Learned Physical Parameters vs. Ground Truth. The parameters learned by our model (colored bars) are compared against the ground-truth values (dashed lines) under varying noise levels ($\delta$).

**Quality of the Learned Physical Representation.** Strong predictive performance should stem from an accurate underlying state representation. We validate this by evaluating the static encoding quality and the model's ability to recover the true physical parameters of the simulation.

Table 1 presents the static encoding RMSE, measuring how accurately each model can infer the physical state from a single observation. The results strongly correlate with the predictive findings. Our quantized extrinsic PIWM achieves the highest encoding accuracy, confirming that its superior representation is the foundation of its predictive power. The intrinsic continuous models, while competitive, exhibit higher encoding error, consistent with their weaker predictive performance.

Finally, we assess whether the dynamics model can learn the true physical parameters (e.g., car length, pole mass) within the learned latent space. As reported in Figure 3, PIWM successfully recovers the ground-truth parameters with low relative error across all environments. This provides

direct evidence that our framework learns a genuinely interpretable representation that is not merely correlated with the physics but is structured in a way that is consistent with them.

Table 1: Static Encoding RMSE (mean $\pm$ std) Comparison of MSE and KL Losses

| Case | Model | $\delta = 0\%$ | | $\delta = 5\%$ | | $\delta = 10\%$ | |
|------|-------|-----|-----|-----|-----|-----|-----|
| | | MSE | KL | MSE | KL | MSE | KL |
| Donkey Car | Intrinsic (Cont.) | $0.13 \pm 0.05$ | $0.45 \pm 0.06$ | $0.16 \pm 0.06$ | $0.52 \pm 0.07$ | $0.20 \pm 0.07$ | $0.58 \pm 0.08$ |
| | Intrinsic (Disc.) | $0.18 \pm 0.04$ | $0.42 \pm 0.05$ | $0.21 \pm 0.05$ | $0.55 \pm 0.06$ | $0.22 \pm 0.06$ | $0.64 \pm 0.07$ |
| | Extrinsic (Cont.) | $0.11 \pm 0.05$ | $0.24 \pm 0.04$ | $0.15 \pm 0.04$ | $0.28 \pm 0.05$ | $0.16 \pm 0.05$ | $0.33 \pm 0.06$ |
| | **Extrinsic (Disc.)** | $0.03 \pm 0.02$ | $0.19 \pm 0.03$ | $0.05 \pm 0.03$ | $0.23 \pm 0.04$ | $0.06 \pm 0.04$ | $0.28 \pm 0.05$ |
| Lunar Lander | Intrinsic (Cont.) | $0.07 \pm 0.05$ | $0.56 \pm 0.26$ | $0.08 \pm 0.15$ | $0.58 \pm 0.37$ | $0.21 \pm 0.07$ | $0.77 \pm 0.38$ |
| | Intrinsic (Disc.) | $0.06 \pm 0.04$ | $0.44 \pm 0.11$ | $0.03 \pm 0.05$ | $0.53 \pm 0.12$ | $0.16 \pm 0.06$ | $0.64 \pm 0.17$ |
| | Extrinsic (Cont.) | $0.11 \pm 0.07$ | $0.33 \pm 0.14$ | $0.14 \pm 0.04$ | $0.39 \pm 0.16$ | $0.15 \pm 0.05$ | $0.45 \pm 0.26$ |
| | **Extrinsic (Disc.)** | $0.03 \pm 0.02$ | $0.24 \pm 0.14$ | $0.09 \pm 0.03$ | $0.29 \pm 0.11$ | $0.12 \pm 0.04$ | $0.35 \pm 0.23$ |

**Analysis and Discussion.** Our experiment demonstrates that the extrinsic architecture with a discrete latent space is optimal for learning physically interpretable world models from weak supervision. This approach achieves superior prediction accuracy (Figure 2) by decoupling perception from physical state abstraction and leveraging quantization as a powerful regularizer against visual noise. The learned representations are not only predictive but also genuinely physically grounded, as evidenced by the model's ability to recover true system parameters (Figure 3) and generate qualitatively plausible visual rollouts that far exceed baseline performance (Figure 4). Crucially, the substantial gains in interpretability and prediction accuracy come at a minimal cost to downstream controller performance (Table 2, Appendix), validating our architecture as a robust and practical solution.

While our framework is effective, future work should focus on scaling its representational capacity and enhancing its use of supervision. For complex, open-world scenarios like autonomous driving, our approach could be extended from predicting simple state vectors to building structured world representations, such as dynamic 3D occupancy grids, where physical priors can be applied to multiple agents. Furthermore, the rich temporal nature of weak supervision signals is currently underutilized. Future methods could process sequences of noisy supervisory signals using filtering or sequence modeling techniques to produce a more refined, temporally coherent learning target, thereby improving the model's robustness and accuracy.

## 5 RELATED WORK

**Trajectory Prediction.** Predicting trajectories is critical for safe planning and control (Fridovich-Keil et al., 2020), but existing methods present trade-offs. While approaches like Hamilton-Jacobi (HJ) reachability offer formal guarantees (Li et al., 2021; Nakamura & Bansal, 2023), they are computationally expensive for online settings. Conversely, mainstream deep learning models are powerful but often rely on handcrafted scene representations (Salzmann et al., 2020) or high-precision maps (Itkina & Kochenderfer, 2023; Hsu et al., 2023), and typically do not produce physically interpretable predictions (Lu et al., 2024; Lindemann et al., 2023; Ruchkin et al., 2022). In contrast, our approach learns directly from raw images with distribution-based weak supervision without requiring handcrafted inputs or goal conditioning.

**Representation Learning.** A central challenge in learning from high-dimensional sequences is creating compact and meaningful latent representations (Shi et al., 2015; Bai et al., 2018). While Variational Autoencoders (VAEs) (Kingma, 2013) are foundational, their standard form learns unstructured latents. A significant line of research attempts to impose structure, either by encouraging disentanglement in continuous spaces with methods like $\beta$-VAE, FactorVAE, and TCVAE (Higgins et al., 2017; Kim & Mnih, 2018; Chen et al., 2018), or by enforcing causality (Yang et al., 2021). However, these methods often fall short of ensuring a direct correspondence with real-world physical states or require precisely labeled data. An alternative is to impose structure via discretization with Vector-Quantized VAEs (VQ-VAEs) and their extensions (Van Den Oord et al., 2017; Razavi et al., 2019; Xue et al., 2019). While effective, their application to physical prediction has been limited due to abstract latent codes and reliance on exact supervision. Even in robotic applications like DVQ-VAE that model structured systems, encoding external environmental factors remains a challenge (Zhao et al., 2024). GOKU-net constrains variables to plausible physical ranges but does not tie them to specific, interpretable quantities (Linial et al., 2021).

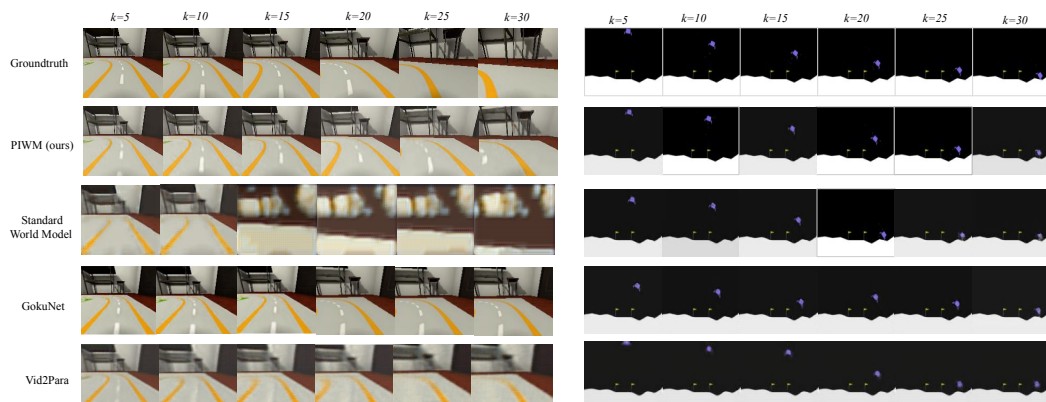

Figure 4: Qualitative Comparison of 30-Step Trajectory Rollouts under $\delta = 5\%$.

Other works incorporate structured priors to enhance learning. Object-centric world models like FOCUS and foundation world models improve efficiency by structuring latents around discrete entities (Ferraro et al., 2023; Mao et al., 2024a). Models like SPARTAN provide outputs that are interpretable by construction but do not incorporate known physical dynamics or action-conditioned prediction (Lei et al., 2024). Priors can also be introduced as task-related rules in Bayesian neural networks (Sam et al., 2024) or through human and self-supervision to guide abstraction (Fu et al., 2021; Wen et al., 2023; Konidaris et al., 2018; Peng et al., 2024; Chen et al., 2022). These approaches, however, often yield abstract representations that lack physical grounding without strong external priors. Closer to our work, Deep Variational Bayes Filters (DVBFs) extend VAEs with a latent dynamics module (Karl et al., 2016). Yet, without explicit physical supervision, they often fail to recover interpretable variables—a limitation our work addresses by leveraging weak supervision.

**World Models.** Classical world models like Dreamer (Hafner et al., 2020) and DayDreamer (Wu et al., 2023) excel at learning from experience for policy learning, but their latent representations are typically uninterpretable (Peper et al., 2025). Many approaches seek to improve physical grounding by incorporating priors, such as bounds on states and actions (Tumu et al., 2023; Sridhar et al., 2023), physics-aware loss functions (Djeumou et al., 2023), or kinematics-inspired layers (Cui et al., 2020). However, these methods are often designed for low-dimensional systems and do not scale well to learning from noisy, high-dimensional images. Other physics-informed methods leverage differential equations to stabilize learning (Zhong & Meidani, 2023; Linial et al., 2021), with frameworks like Phy-Taylor using Taylor monomials to structure the latent dynamics (Mao et al., 2025a). Approaches like sparse identification and differentiable physics require access to the underlying state variables and are not designed to learn from raw visual inputs (Yao et al., 2024; Brunton et al., 2016; de Avila Belbute-Peres et al., 2018).

Recent advances have produced powerful but distinct world models. For instance, 3D occupancy-based models improve forecasting but their internal states are not explicitly aligned with physical variables (Zheng et al., 2024; Min et al., 2023; Yan et al., 2024; Zuo et al., 2024). Concurrently, neuro-symbolic models enhance generalization but require predefined symbolic inputs not available from raw sensor data (Balloch et al., 2023; Liang et al., 2024). Our approach is distinct from these paradigms as it learns physically interpretable representations directly from images using weak supervision. This also contrasts with the most closely related work, Vid2Param (Asenov et al., 2019), which requires full supervision and struggles with dynamics prediction.

## 6   CONCLUSION AND BROADER IMPACT

We presented the Physically Interpretable World Model, a framework that learns physically-grounded latent representations from images using only weak distributional supervision. Our systematic evaluation demonstrates that an extrinsic architecture with a discrete latent space yields accurate and robust predictions and successfully recovers the system's true physical parameters. This work not only provides direct evidence of a genuinely interpretable model but also offers a practical path toward more trustworthy and reliable autonomous systems in high-stakes applications.

## ETHICS STATEMENT

This work does not involve human subjects, sensitive personal data, or potentially harmful applications. No ethical concerns regarding discrimination, bias, privacy, or security arise in the scope of this research. All experiments were conducted using publicly available datasets under appropriate licenses, and all methods follow the ICLR Code of Ethics.

## REPRODUCIBILITY STATEMENT

To ensure reproducibility, we provide implementation details in the supplementary materials, including code. The main paper describes the model architecture and experimental settings, while additional training details are documented in the appendix. Anonymous source code is included as supplementary materials. With the provided resources.

## USE OF LARGE LANGUAGE MODELS (LLMS)

Large language models were only used for language polishing and code debugging. They did not contribute to research ideation, experimental design, or the writing of scientific content.

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

APPENDIX

A. TRAINING ALGORITHMS

The training procedures for the ablation and proposed models are summarized in Algorithm 2 and Algorithm 1, respectively.

For the PIWM model (Algorithm 1), observations are first encoded into discrete latent representations using a Vision VQ-VAE module. These discrete latents are further mapped into interpretable physical states via a physical encoder $\mathcal{E}_p$. The dynamics model $\phi_\theta$ predicts the future physical state based on the current and next physical states. The predicted physical state is decoded back into the latent space and finally reconstructed into the future observation. The loss is computed similarly, with both reconstruction and weak supervision terms.

For the ablation model (Algorithm 2), consecutive observations are first encoded using the encoder $\mathcal{E}$ into latent variables. The dynamics model $\phi_\theta$ predicts the future latent state by taking as input the latent representations from two consecutive frames. The decoder $\mathcal{D}$ reconstructs the observation at the future time step, and a reconstruction loss is computed. In addition, a weak supervision loss is applied to regularize the encoded latents based on interval labels.

The main difference between the ablation and proposed models lies in the latent structure and dynamics modeling: the ablation model operates directly in the visual latent space, while the proposed model enforces a structured, interpretable physical latent space for dynamics prediction.

We further include additional data-driven ablation variants, where the dynamics predictor $\phi_\theta$ in the ablation model is replaced by a sequential model, such as an LSTM or Transformer network. These models are trained to predict the next latent state given the previous two latent representations. The rest of the architecture (encoder, decoder, and loss computation) remains identical to the basic ablation setup.

The inference procedure is similar across all variants, using a sequence of encoded observations to roll out future predictions recursively over multiple horizons.

B. BACKPROPAGATION THROUGH DYNAMICS

To optimize the parameters $\theta$ in the structured dynamics model $\phi(\cdot; \theta)$, we compute gradients of the loss using the chain rule. The loss may depend on the predicted interpretable state $\hat{x}_{t+2}$ (e.g., prediction error or safety violation), or on the sequence $\{x'_t, x'_{t+1}, x'_{t+2}\}$ for temporal consistency. We compute the gradient as:

$$\frac{\partial \mathcal{L}}{\partial \theta} = \frac{\partial \mathcal{L}}{\partial \hat{x}_{t+2}} \cdot \frac{\partial \hat{x}_{t+2}}{\partial \theta}.$$

Since $\hat{x}_{t+2}$ is computed as $\phi(x'_{t+1}, a_{t+1}; \theta)$, which in turn depends on previous states through recursive application of $\phi$, we expand the gradient recursively:

$$\frac{\partial \hat{x}_{t+2}}{\partial \theta} = \frac{\partial \phi(x'_{t+1}, a_{t+1}; \theta)}{\partial \theta} + \frac{\partial \phi(x'_{t+1}, a_{t+1}; \theta)}{\partial x'_{t+1}} \cdot \frac{\partial x'_{t+1}}{\partial \theta}.$$

This recursion can be unrolled through multiple time steps, enabling gradients to propagate through interpretable state sequences and support end-to-end learning of both state representations and dynamics parameters.

C. EXPERIMENT SETUP

We evaluate our approach on three control benchmarks of increasing complexity: CartPole, Lunar Lander, and Donkey Car. These tasks differ in state dimensionality, control difficulty, and underlying dynamics. CartPole requires balancing a pole on a moving cart using discrete left/right forces. The system has a four-dimensional state space (position, velocity, angle, angular velocity) and a binary discrete action space. It includes four dynamics parameters: cart mass, pole mass, pole length, and applied force magnitude. Lunar Lander involves landing a spacecraft on flat terrain using main and side engines. It has an eight-dimensional state space (positions, velocities, angle,

---

**Algorithm 1:** Training PIWMs

---

**Input:** Training set $\mathcal{S} = \{(y_{1:M}, a_{1:M}, [x^{\text{lo}}_{1:M}, x^{\text{up}}_{1:M}])\}^N_{i=1}$, batch size $B$, weight $\lambda$, optimizer
      `Adam()`
**Output:** Trained parameters $w$ for $(\mathcal{E}, \mathcal{E}_p, \phi_\theta, \mathcal{D}_p, \mathcal{D})$

1   Initialize $w$ randomly;
2   **for** $i \leftarrow 1$ **to** $|\mathcal{S}|/B$ **do**
3      Sample a batch of $B$ sequences from $\mathcal{S}$;
4      **foreach** *sequence* $(y_{1:M}, a_{1:M}, [x^{lo}_{1:M}, x^{up}_{1:M}])$ *in batch* **do**
5         **for** $t \leftarrow 1$ **to** $M-2$ **do**
6           $z_t \leftarrow \mathcal{E}(y_t), \quad z_{t+1} \leftarrow \mathcal{E}(y_{t+1})$;
7           $z^*_t \leftarrow \text{Quantize}(z_t), \quad z^*_{t+1} \leftarrow \text{Quantize}(z_{t+1})$;
8           $\hat{x}'_t \leftarrow \mathcal{E}_p(z^*_t), \quad \hat{x}'_{t+1} \leftarrow \mathcal{E}_p(z^*_{t+1})$;
9           $a_t \leftarrow h(y_t)$;
10          $\hat{x}_{t+2} \leftarrow \phi_\theta(\hat{x}'_t, \hat{x}'_{t+1}, a_t)$;
11          $\hat{z}^*_{t+2} \leftarrow \mathcal{D}_p(\hat{x}_{t+2})$;
12          $\hat{y}_{t+2} \leftarrow \mathcal{D}(\mathbf{e}_{\hat{z}^*_{t+2}})$;
13          Sample $\tilde{x}_t \sim \mathcal{U}(x^{lo}_t, x^{up}_t)$; ;            // Weak supervision sample
14          Sample $\tilde{x}_{t+2} \sim \mathcal{U}(x^{lo}_{t+2}, x^{up}_{t+2})$;
15          Compute reconstruction loss: $\mathcal{L}_{\text{rec}} = \|y_{t+2} - \hat{y}_{t+2}\|^2_2$;
16          Compute state supervision loss: $\mathcal{L}_{\text{state}} = \|\hat{x}'_t - \tilde{x}_t\|^2_2$;
17          Compute dynamics loss: $\mathcal{L}_{\text{dyn}} = \|\hat{x}_{t+2} - \tilde{x}_{t+2}\|^2_2$;
18          Compute latent consistency loss: $\mathcal{L}_{\text{latent}} = \text{CrossEntropy}(\hat{z}^*_{t+2}, z^*_{t+2})$;
19          Total loss: $\mathcal{L} = \mathcal{L}_{\text{rec}} + \lambda(\mathcal{L}_{\text{state}} + \mathcal{L}_{\text{latent}} + \mathcal{L}_{\text{dyn}})$;
20      Update parameters: $w \leftarrow \text{Adam}(\theta, \nabla_w \mathcal{L})$;
21   **return** $w$

---

angular velocity, contact flags) and a four-dimensional continuous action space for engine thrust. Two key parameters govern the dynamics: main engine power and side engine power. Donkey Car simulates autonomous vehicle control with continuous throttle and steering inputs. We adopt a bicycle dynamics model, where the primary physical parameter to learn is the effective car length. This task presents additional complexity due to nonlinear dynamics and tight coupling between steering and acceleration.

The vision encoder uses two convolutional layers followed by a channel projection. The latent space has 64 dimensions and is quantized using a codebook with 512 entries. The total loss includes reconstruction loss, codebook update loss, and a commitment loss weighted by a factor of 0.25. The interpretable encoder is implemented as a 2-layer Transformer with 4 attention heads and a feedforward dimension of 512. Each codebook index is embedded into a 16-dimensional vector. The encoder output is mean-pooled and passed through a linear layer to regress physical states. The decoder mirrors the encoder and predicts discrete latent indices, optimized with a cross-entropy loss. The total loss includes state regression loss and index reconstruction loss.

## D. INTERPRETABLE CODEBOOK IN VQVAE

For our intrinsic architecture, we aimed to design a VQ-VAE where the discrete codebook itself would be physically interpretable, allowing a single encoder to map directly from images to a structured, meaningful latent space. To this end, we explored several variants.

Our first attempt involved making the physical dimensions of the codebook learnable. In this configuration, a partition of each codebook vector was initialized with values from a discretized grid of physical states, but these values were allowed to be updated via backpropagation during end-to-end training. We hypothesized that the model could learn an optimal embedding for the physical states. However, this approach proved to be unstable. Under the guidance of only weak supervision, the values in these dimensions would often drift significantly from their intended physical semantics, leading to a failure to learn a coherent physical representation.

To enforce a stronger semantic structure and encourage disentanglement, we next designed concept-specific codebooks. This approach assigned separate, independent embedding tables to different physical variables (e.g., one codebook for position, another for angle). These models were trained with modified loss functions that incorporated regularization terms to encourage semantic alignment and penalize mismatches between predicted states and codebook semantics. Despite this stronger structural prior, all variants still suffered from severe codebook utilization collapse, where the model would rely on only a very small subset of the available codebook entries during both training and testing. This resulted in poor diversity in the latent representations and a failure to capture the full range of physical states.

We attribute the failure of these explorations to the lack of a sufficiently strong signal from the weak supervision to guide such a complex and under-constrained optimization problem. Although we adjusted the commitment loss and interpretability regularization weights, the issue persisted.

Given the limitations of these more flexible, learnable designs, we ultimately adopted the simpler and more constrained approach for the intrinsic-discrete model described in Section 3.1, which yielded better and more stable results. In that architecture, we partition each codebook vector $\mathbf{e}_k$ into a physical part $\mathbf{e}_k^p$ and a visual part $\mathbf{e}_k^v$, but the physical part $\mathbf{e}_k^p$ is fixed as a non-learnable constant representing a specific point on the physical state grid. This method, while sacrificing the flexibility of a learnable physical embedding, proved far more robust against codebook collapse and provided the necessary stability for the model to learn effectively.

E. ADDITIONAL RESULTS

This section provides supplementary results that further validate the conclusions presented in the main paper. We include detailed prediction performance for the CartPole environment, an analysis of the learned physical parameters for CartPole.

Figure 5 displays the 30-step prediction Root Mean Square Error (RMSE) for the CartPole environment. The results are consistent with the findings from the Donkey Car and Lunar Lander environments discussed in the main text. The extrinsic models, particularly our quantized (discrete) variant, consistently achieve lower prediction error across all noise levels ($\delta$) compared to intrinsic models and data-driven baselines like LSTM and Transformer. This reinforces our conclusion that decoupling perception from physical state inference through an extrinsic architecture provides superior long-horizon prediction stability.

To further demonstrate that our model learns a genuinely interpretable representation, we evaluated its ability to recover the true physical parameters of the CartPole simulation. As shown in Figure 6, our PIWM framework successfully identifies the ground-truth values for the pole's mass, the pole's length, the cart's length, and the applied force with high accuracy, especially under low noise conditions ($\delta$=0). While the variance of the learned parameters increases with higher levels of supervision noise, the model's estimates remain centered around the true values, providing strong evidence that the latent space is structured in a physically meaningful way.

Table 2: Controller Performance on Reconstructed Observations Across All Variants and Noise Levels

| Case | Input Type | Latent Space | Supervision Noise Level ($\delta$) | | |
| --- | --- | --- | --- | --- | --- |
| | | | 0% | 5% | 10% |
| Donkey Car (Action RMSE ↓) | One-Stage (Intrinsic) | Continuous | $0.12 \pm 0.04$ | $0.13 \pm 0.04$ | $0.15 \pm 0.05$ |
| | | Discrete | $0.21 \pm 0.15$ | $0.29 \pm 0.16$ | $0.32 \pm 0.20$ |
| | Two-Stage (Extrinsic) | Continuous | $0.15 \pm 0.05$ | $0.16 \pm 0.05$ | $0.22 \pm 0.06$ |
| | | Discrete | $0.15 \pm 0.04$ | $0.17 \pm 0.05$ | $0.19 \pm 0.05$ |
| Lunar Lander (Action Acc. ↑) | One-Stage (Intrinsic) | Continuous | $93.0\% \pm 1.8\%$ | $90.5\% \pm 2.0\%$ | $87.1\% \pm 2.2\%$ |
| | | Discrete | $85.5\% \pm 2.5\%$ | $82.1\% \pm 2.8\%$ | $78.3\% \pm 3.1\%$ |
| | Two-Stage (Extrinsic) | Continuous | $86.2\% \pm 2.4\%$ | $83.5\% \pm 2.6\%$ | $80.0\% \pm 2.9\%$ |
| | | Discrete | $91.5\% \pm 2.1\%$ | $88.6\% \pm 2.3\%$ | $84.5\% \pm 2.5\%$ |
| Cart Pole (Action Acc. ↑) | One-Stage (Intrinsic) | Continuous | $98.0\% \pm 1.0\%$ | $96.5\% \pm 1.2\%$ | $94.0\% \pm 1.5\%$ |
| | | Discrete | $95.0\% \pm 1.6\%$ | $91.5\% \pm 2.0\%$ | $87.2\% \pm 2.5\%$ |
| | Two-Stage (Extrinsic) | Continuous | $95.5\% \pm 1.5\%$ | $92.0\% \pm 1.8\%$ | $88.0\% \pm 2.2\%$ |
| | | Discrete | $97.2\% \pm 1.1\%$ | $95.0\% \pm 1.4\%$ | $92.5\% \pm 1.8\%$ |

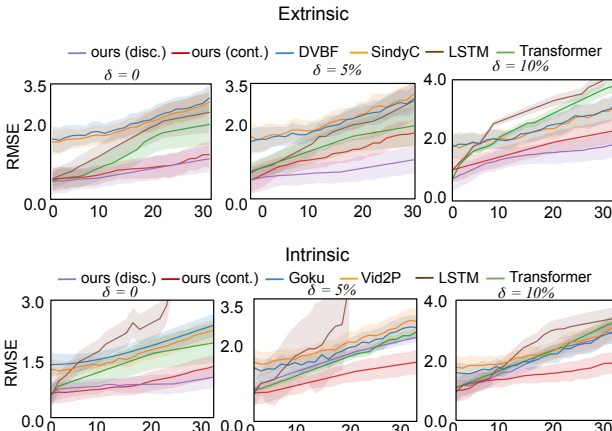

Figure 5: Prediction performance of CartPole. The Root Mean Square Error (RMSE) of our PIWM variants (extrinsic methods, left; intrinsic methods, right) is compared against baselines over a 30-step prediction horizon in the CartPole across varying levels of weak supervision ($\delta$).

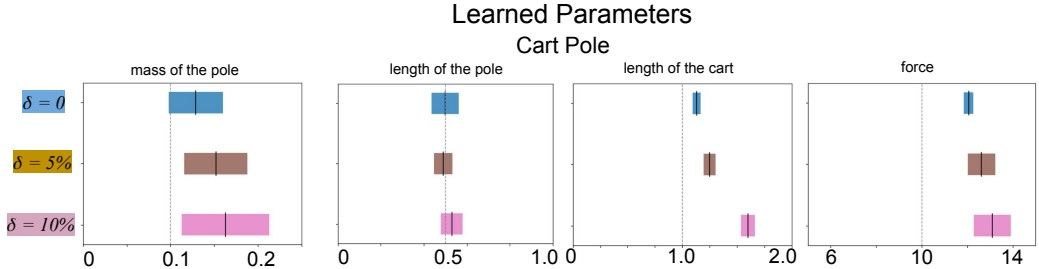

Figure 6: Learned Physical Parameters vs. Ground Truth. The parameters learned by our model (colored bars) are compared against the ground-truth values (dashed lines) under varying noise levels ($\delta$).

## F. DYNAMICS OF CART POLE, LUNAR LANDER, AND DONKEY CAR

Algorithms 3, 4, and 5 describe the dynamics of the Cart Pole, Lunar Lander, and Donkey Car environments, respectively.

Algorithm 3 models the Cart Pole system, capturing the relationships between the cart's horizontal motion and the pole's angular motion. The dynamics incorporate applied forces and resulting accelerations, governing how the position, velocity, pole angle, and angular velocity evolve over time.

Algorithm 4 presents the dynamics of the Lunar Lander, where discrete actions determine the firing of the main and side engines. The equations govern the lander's position, velocity, orientation, and angular velocity, enabling simulation of its flight behavior under various control inputs.

Algorithm 5 introduces a simplified learnable bicycle model used to approximate the Donkey Car's dynamics. While the true Donkey Car system implemented in Unity involves complex physics and interactions, this model captures essential relationships between the vehicle's position, heading, and speed under steering and acceleration commands, allowing efficient learning and prediction without requiring access to the full simulation engine.

---

**Algorithm 2:** Ablation Model: Learning Dynamics from Consecutive Latents to Predict Future States

---

**Input:** Training dataset $V$ containing sequences of images and weak state labels $(y_{1:M}, [x_{1:M}^{lo}, x_{1:M}^{up}])$, batch size $B$, weight $\lambda_1$, optimizer `Adam()`

**Output:** Trained model $(\mathcal{E}, \phi_\theta, \mathcal{D})$ with parameters $\theta$

**1** Initialize parameters $\theta$ randomly;

**2 for** $i \leftarrow 1$ **to** $len(V)/B$ **do**

**3**     Sample a batch of $B$ sequences from $V$;

**4**     **foreach** *sequence in batch* **do**

**5**        **for** $j \leftarrow 1$ **to** $M - 2$ **do**

**6**          $(\mu_{z_j}, \sigma_{z_j}^2) = \mathcal{E}(y_j)$;

**7**          $(\mu_{z_{j+1}}, \sigma_{z_{j+1}}^2) = \mathcal{E}(y_{j+1})$;

           ;         // Encode two consecutive observations

**8**          $z_j \sim \mathcal{N}(\mu_{z_j}, \sigma_{z_j}^2)$;

**9**          $z_{j+1} \sim \mathcal{N}(\mu_{z_{j+1}}, \sigma_{z_{j+1}}^2)$;

           ;         // Sample latent variables

**10**         $\Delta\hat{z}_j = \phi_\theta(z_j, z_{j+1})$;

           ;         // Predict latent dynamics based on $z_j$ and $z_{j+1}$

**11**         $\hat{z}_{j+2} = z_{j+1} + \Delta\hat{z}_j$;

           ;         // Predict next latent state

**12**         $\hat{y}_{j+2} = \mathcal{D}(\hat{z}_{j+2})$;

           ;         // Decode predicted latent to reconstruct future observation

**13**         $\mathcal{L}_0 = \frac{1}{D} \sum_{k=1}^{D} \|\hat{y}_{j+2} - y_{j+2}\|^2$;

           ;         // Reconstruction loss against true $y_{j+2}$

**14**         $p \sim \mathcal{N}\left( \frac{x_j^{up} - x_j^{lo}}{2}, \left( \frac{x_j^{lo} + x_j^{up}}{6} \right)^2 \right)$;

**15**         $q \sim \mathcal{N}(\mu_{z_j}, \sigma_{z_j}^2)$;

**16**         $\mathcal{L}_1 = \sum_{k=1}^{d} \left\| \frac{1}{2}\left( \frac{\sigma_p^2}{\sigma_q^2} + \frac{(\mu_q - \mu_p)^2}{\sigma_q^2} - 1 + \ln\frac{\sigma_q^2}{\sigma_p^2} \right) \right\|^2$;

           ;         // KL divergence with weak supervision

**17**     Update parameters: $\theta \leftarrow \text{Adam}(\theta, \nabla_\theta(\mathcal{L}_0 + \lambda_1 \mathcal{L}_1))$;

**18 return** $\theta$

---

---

**Algorithm 3:** Dynamics of Cart Pole

**Input:** Current state $z = [x, \dot{x}, \theta, \dot{\theta}]$, action $a$

**Output:** Updated state $z_{\text{new}} = [x_{\text{new}}, \dot{x}_{\text{new}}, \theta_{\text{new}}, \dot{\theta}_{\text{new}}]$

// Extract State Variables

1 $x, \dot{x}, \theta, \dot{\theta} \leftarrow z[:, 0], z[:, 1], z[:, 2], z[:, 3]$

// Convert Action to Force

2 $F \leftarrow \text{force\_mag} \times (2 \cdot a - 1)$

// Compute Trigonometric Values

3 $\cos(\theta) \leftarrow \text{costheta}, \quad \sin(\theta) \leftarrow \text{sintheta}$

// Compute Intermediate Variable

4 $\text{temp} \leftarrow \frac{F + m_p \cdot l \cdot \dot{\theta}^2 \cdot \sin(\theta)}{m_p + m_c}$

// Calculate Angular Acceleration

5 $\ddot{\theta} \leftarrow \frac{g \cdot \sin(\theta) - \cos(\theta) \cdot \text{temp}}{l \cdot \left( \frac{4}{3} - \frac{m_p \cdot \cos^2(\theta)}{m_p + m_c} \right)}$

// Calculate Linear Acceleration

6 $\ddot{x} \leftarrow \text{temp} - \frac{m_p \cdot l \cdot \ddot{\theta} \cdot \cos(\theta)}{m_p + m_c}$

// Update State Variables

7 $x_{\text{new}} \leftarrow x + \tau \cdot \dot{x} \; \dot{x}_{\text{new}} \leftarrow \dot{x} + \tau \cdot \ddot{x} \; \theta_{\text{new}} \leftarrow \theta + \tau \cdot \dot{\theta} \; \dot{\theta}_{\text{new}} \leftarrow \dot{\theta} + \tau \cdot \ddot{\theta}$

// Return Updated State

8 $z_{\text{new}} \leftarrow [x_{\text{new}}, \dot{x}_{\text{new}}, \theta_{\text{new}}, \dot{\theta}_{\text{new}}]$ **return** $z_{\text{new}}$;

---

---

**Algorithm 4:** Dynamics of Lunar Lander

**Input:** Current states $\mathbf{s} = [x, y, \dot{x}, \dot{y}, \theta, \dot{\theta}]$, actions $\mathbf{a}$ (0: do nothing, 1: fire left, 2: fire main, 3: fire right)

**Output:** Updated states $\mathbf{s}_{\text{new}} = [x_{\text{new}}, y_{\text{new}}, \dot{x}_{\text{new}}, \dot{y}_{\text{new}}, \theta_{\text{new}}, \dot{\theta}_{\text{new}}]$

// Unpack State Variables

1 $x \leftarrow \mathbf{s}[:, 0], \quad y \leftarrow \mathbf{s}[:, 1] \; \dot{x} \leftarrow \mathbf{s}[:, 2], \quad \dot{y} \leftarrow \mathbf{s}[:, 3] \; \theta \leftarrow \mathbf{s}[:, 4], \quad \dot{\theta} \leftarrow \mathbf{s}[:, 5]$

// Calculate Engine Direction and Dispersion

2 $\text{tip} \leftarrow [\sin(\theta), \cos(\theta)] \; \text{side} \leftarrow [-\cos(\theta), \sin(\theta)]$

// Process Actions

3 $\text{fire\_main} \leftarrow (\mathbf{a} == 2) \; \text{fire\_left} \leftarrow (\mathbf{a} == 1) \; \text{fire\_right} \leftarrow (\mathbf{a} == 3)$

// Compute Main Engine Thrust

4 $m_{\text{power}} \leftarrow \text{fire\_main} \; \dot{x} \leftarrow \dot{x} - \text{tip}[:, 0] \cdot \text{main\_power} \cdot m_{\text{power}}/\text{FPS}$
$\dot{y} \leftarrow \dot{y} + \text{tip}[:, 1] \cdot \text{main\_power} \cdot m_{\text{power}}/\text{FPS}$

// Compute Side Engine Thrust

5 $s_{\text{power}} \leftarrow \text{fire\_left} + \text{fire\_right} \; \text{direction} \leftarrow \text{fire\_right} - \text{fire\_left}$
$\dot{x} \leftarrow \dot{x} + \text{side}[:, 0] \cdot \text{side\_power} \cdot s_{\text{power}} \cdot \text{direction}/\text{FPS}$
$\dot{\theta} \leftarrow \dot{\theta} + \text{side\_power} \cdot s_{\text{power}} \cdot \text{direction}/\text{FPS}$

// Update Position and Angle

6 $x \leftarrow x + \dot{x}/\text{FPS} \; y \leftarrow y + \dot{y}/\text{FPS} \; \theta \leftarrow \theta + \dot{\theta}/\text{FPS}$

// Create Updated States

7 $\mathbf{s}_{\text{new}} \leftarrow [x, y, \dot{x}, \dot{y}, \theta, \dot{\theta}]$ **return** $\mathbf{s}_{\text{new}}$;

---

---

**Algorithm 5:** Dynamics of Bicycle Model

---

**Input:** Current state $s = [x, y, \theta, v]$, action $a = [\delta, a_{\text{acc}}]$
**Output:** Updated state $s_{\text{new}} = [x_{\text{new}}, y_{\text{new}}, \theta_{\text{new}}, v_{\text{new}}]$
// Extract State Variables
1   $x, y, \theta, v \leftarrow s[:, 0], s[:, 1], s[:, 2], s[:, 3]$
// Extract Action Variables
2   $\delta, a_{\text{acc}} \leftarrow a[:, 0], a[:, 1]$
// Update State Variables
3   $x_{\text{new}} \leftarrow x + v \cdot \cos(\theta) \cdot \tau \; y_{\text{new}} \leftarrow y + v \cdot \sin(\theta) \cdot \tau \; \theta_{\text{new}} \leftarrow \theta + \frac{v}{L} \cdot \tan(\delta) \cdot \tau \; v_{\text{new}} \leftarrow v + a_{\text{acc}} \cdot \tau$
// Return Updated State
4   $s_{\text{new}} \leftarrow [x_{\text{new}}, y_{\text{new}}, \theta_{\text{new}}, v_{\text{new}}]$ **return** $s_{\text{new}}$;

---

