# OpenReview forum: "Can Weak Quantization Make World Models Physically Interpretable?"
_ICLR.cc/2026/Conference — ICLR 2026 Conference Withdrawn Submission_

### Official Review · Reviewer_4jhJ · 2025-10-28

**Soundness:** 3
**Presentation:** 3
**Contribution:** 3
**Rating:** 4
**Confidence:** 3

**Summary:**

The paper proposes PIWM, which learn physics-aligned latent spaces from images using only weak supervision. It combines a visual autoencoder with a physics-based dynamics model that learns unknown parameters like friction or mass. Two designs are tested, first intrinsic (joint learning) and then extrinsic (two-stage), with the extrinsic version proving more stable and interpretable. Experiments on simple control tasks show that PIWM can make physically consistent predictions without full state supervision.

**Strengths:**

- I liked the ablation study. it’s clear that authors test different design choices like intrinsic vs. extrinsic and continuous vs. discrete setups. It helps justify the final model design instead of just showing one result.

- The paper is well written and easy to follow. Including the algorithms in the appendix was a good idea and it makes the training process easier to understand.

- The paper provides the code along with instructions, which is a good move. It helps with reproducibility and is appreciated by the community.

- It’s good that the physics loss is reported separately. It clearly shows whether the model is actually learning the physical part. The reconstruction visualizations are also helpful to confirm that both the visual and physical parts are trained properly.

**Weaknesses:**

- The paper lacks an explicit probabilistic formulation of the overall generative model. Unlike typical probabilistic world models that define a full joint likelihood/generative process, and then deriving approximations (e.g., through an ELBO either in theory or clearly stating that). Please see for example  eq. (3) and (6) in PlaNet paper.
In your model you present individual training losses without connecting them to a unified evidence objective (which are somehow direct Monte-Carlo approximation of your dismantled ELBO terms).

- The paper does not clearly explain the training of the controller $h$. It remains unclear whether the controller is pretrained, jointly optimized with the encoders, or learned during the dynamics phase. Since the controller’s outputs $a$ directly affect the prediction and dynamics learning, the lack of detail on its training stage and objective makes it hard to follow up.

- The exps mainly focus on Markovian systems, where transitions depend only on the most recent state and action. As a result, it remains unclear how the proposed model behaves under non-Markovian, where temporal dependencies extend beyond one step. Evaluating the model on a system with delayed effects could be a good practice to show how it generalizes (or if you believe this is not feasible in the current shape of your work, mention it as a limitation)


- All evaluated exps rely on single-mode, continuous dynamics, and the model does not appear to address systems with multiple discrete or hybrid modes.  The paper would benefit from either (i) an additional experiment demonstrating the model’s behavior under multi-modal dynamics (for example a bouncing ball in gravity room, or NASCAR style dataset with multiple mode switches) or (ii) a clear discussion of how the current one could be extended to handle such scenario.

**Questions:**

Thank you for your work and all efforts. I have some questions about the paper:

- You describes the extrinsic representation learning as a two-stage process, that the visual encoder is first trained and then frozen before training the physical encoder and the dynamics model. In your Algorithm 1, it appears to combine all losses including $rec, state, latent$ and $dyn$ into a single joint optimization step. Could the authors clarify whether the dynamics parameters $\theta$ are trained separately using $\mathcal{L}_{dyn}$ or whether all losses are optimized simultaneously?

- Could you clarify whether your model can be expressed as a full probabilistic generative process (for example defining the joint likelihood of states, physical parameters, and observations)? In probabilistic world models, it is common to first formulate the complete generative objective and then describe which components are approximated (e.g., via an ELBO).


- Could the authors clarify how the controller is trained or obtained? Specifically, is it optimized jointly with the visual encoder, trained in a separate stage (e.g., after freezing the recognition or physical autoencoders), or assumed to be a fixed policy learned beforehand (e.g., via imitation or reinforcement learning)?

- The experiments appear to assume single-mode, continuous dynamics (like a smooth Newtonian motion without discrete transitions). How would the proposed PIWM framework behave in a multi-modal setting, where the system dynamics change across discrete modes (e.g. consider a bouncing ball in gravity room, when after each bounce the dynamics change completely)? Can the model infer or represent such mode switches within its latent structure, or would it require an explicit discrete latent variable (as in hybrid or HMM-style models)? If not, how could PIWM be extended to handle such cases?



- The dynamics model is trained using short, three-step temporal windows in your algorithm 1, that assumes the functional form of
$\phi()$ is known. This design choice is fine for systems with simple-univariate analytical dynamics (e.g., low-dimensional control setups) but may not generalize when
$\phi()$ is partially unknown, highly nonlinear, multiple systems seen in the obs, or exhibits delayed dependencies (your states evolve non-Markovian style). Could you discus under what conditions this local identification strategy remains valid and how to extend it to more general cases? For example, through multi-step rollouts, or latent recurrent updates ?
Then what computational costs such extensions would need?

- In lines 862-863: Why do the latent dimensions drift from physical meaning under weak supervision? Is there any intuition for that? Is this because of under-constrained optimization, entanglement with visual features, or the absence of strong temporal or structural regularization?

- In the results, the continuous latent variant performs better under the intrinsic (one-stage) setup, while the discrete variant performs better under the extrinsic (two-stage) setup. Why is that happening? Is it related to the way the physical codebook vectors are hardcoded or constrained in the intrinsic model that may affect generalization, or are there other factors?


---
a few minor issues:

- In line 823 (line 10 algorithm 1), you model your dynamics model parameterized by two previous time steps. In your gradient calculation  (lines 790-800) it is changed to a single time step variable though. Please make your notation consistent.

- In eq (9), L_latent ..., no?

- Please provide details of setting hyperparameters like learning rates, optimization setup, etc. Also please explain how did you come up with such setting.

- Your graphical model is in a single time step. Why not in multiple time steps to visualize the evolution of your world more clearly?

---

I am eager to further strengthen my positive opinion about the paper, depending on the authors response to my queries.

---

---

> ### Author Response · Authors · 2025-11-14
>
> 4jhJ-Q1:Probabilistic formulation
>
>
> The PIWM can be written in the reviewer’s preferred probabilistic style.
> Specifically, the implied generative process factorizes as:
> The implied generative process of PIWM can be written as:
>
>
> $$
> p(y_{1:T}, z_{1:T}, \theta) = \\
>     p(\theta)\,\prod_{t=1}^{T} \\
>     p(z_t \mid z_{t-1}, a_{t-1}, \theta)\,
>     p(y_t \mid z_t).
> $$
>
>
> Here, $z_t$ denotes the physical latent state, $\theta$ collects the unknown physical parameters,
> $p(y_t \mid z_t)$ corresponds to the deterministic decoder, and
> $p(z_t \mid z_{t-1}, a_{t-1}, \theta)$ corresponds to our structured physics-based transition.
>
>
> A Monte-Carlo estimate of the negative log-likelihood decomposes into the following objective:
>
>
> $$
> -\log p(y_{1:T}, z_{1:T}, \theta)
> \;\approx\;
> \underbrace{\|y_t - \hat{y}_t\|}_{\text{reconstruction}}
> \;+\;
> \underbrace{\|z_t - \tilde{z}_t\|}_{\text{weak supervision}}
> \;+\;
> \underbrace{\| z_{t+1} - f_\theta(z_t, a_t) \|}_{\text{physics consistency}}.
> $$
>
>
>
>
> 4jhJ-Q2 “Is controller pretrained?”
>
>
> Yes, the controller is pretrained and fixed during all world-model training.
> It is used solely to generate trajectories with diverse behaviors.
> It is never jointly optimized with the encoder, decoder, or dynamics module.
>
>
> 4jhJ-Q3: “Multi-modal dynamics (bouncing ball, NASCAR)”
>
>
> Our work explicitly focuses on single-mode, continuous ODE-based dynamics, which are the dominant class in robotic control benchmarks. This is an exciting direction, but remains outside the scope of the present work. We will add this as a scopre restriction and discuss how PIWM could be extended using hybrid latent dynamics.
>
>
> 4jhJ-Q4 Are dynamics parameters trained separately or jointly？
>
>
> In the extrinsic model, the dynamics parameters θ are trained separately, after the visual encoder is fully trained and frozen.
> Stage 1: train visual encoder/decoder only
> Freeze visual encoder
> Stage 2: train physical encoder
> Stage 3: train dynamics model
> We will make this separation explicit to avoid confusion.
>
>
> 4jhJ-Q5 “Why do the latent dimensions drift from physical meaning under weak supervision?”
>
>
> This drift arises naturally due to the under-constrained nature of weak supervision.
> Three factors contribute:
> 1. Entanglement with visual reconstruction.
> The autoencoder is simultaneously encouraged to minimize reconstruction error.
> Without a strong constraint, the model uses the “physical” dimensions to encode visual texture, lighting, or perspective features—shifting them away from physical meaning.
> 2. Identifiability gap under weak supervision.
> A weak-interval label (e.g., uniform over ±10% of the state space) provides only a coarse region, not the true value.
> Many different codebook embeddings satisfy this constraint equally well.
> The optimization thus drifts to arbitrary embeddings.
> 3. Lack of structural anchoring without discretization.
> When physical dimensions are learnable and continuous, nothing prevents them from sliding along the manifold that reduces reconstruction loss.
> The discrete version, in contrast, anchors physical semantics to fixed codewords, preventing drift.
> This is exactly why the learnable-physical-codebook version was unstable, and why the discrete-extrinsic model achieves the best consistency.

---

### Official Review · Reviewer_LYWn · 2025-10-31

**Soundness:** 1
**Presentation:** 2
**Contribution:** 2
**Rating:** 2
**Confidence:** 4

**Summary:**

This paper approaches trajectory prediction with physically interpretable latent dynamics models.
Continuous and quantized autoencoders are compared as well as intrinsic (only one AE) and extrinsic approaches (two-staged AE).
Correspondence to physical quantities is enforced on the second AE in the extrinsic variant and on a part of the latent space in the intrinsic variant.
This is done by minimizing the L2 loss of the latent to noisy ground-truth state values.
The parameters for known dynamic equations are also learned.
For these approaches and several baselines, the prediction performance over different horizons is evaluated.
For the proposed models, also the error of predicted physical states and parameters are evaluated.
The results indicate that the quantized extrinsic variant made provide a good trade-off for the evaluated problems.

**Strengths:**

- The evaluation of different encoding approaches into a physical latent state space is interesting.
- The technical description is well written and easy to follow.
- The introduction of the formal notation of the physics-informed world model approach reads well and could be useful as a reference for follow-up works in the world model learning domain.

**Weaknesses:**

Major
- The main issue of this paper is that it uses samples from a distribution p(x) of ground truth state x for training and denotes this a "weak supervision", while it is clearly standard strong supervision. The noise model is not described in detail. Several experiments are also performed with noise level 0, meaning that actual ground truth values are used for training. While this dependency of the performance wrt to noise levels is interesting, the paper is clearly overstating by claiming weak supervision. Please revise!

- Dynamics model: It is not clear to me how the structured dynamics model works in detail.
  For example, does it only propagate the physical state or also the the visual latent?
  Also assuming knowledge of ground truth dynamics (with only missing parameters) is a very strong assumption.
  The state is not Markov. Instead, the state derivatives seem to be computed explicitly by finite differences to propagate the state using the manually designed physics-based dynamics model. Why not include state derivatives in the model and let the encoder learn to predict them too? This would require a recurrent encoder or multiple frames as input though. Please comment.
  Crucially, it seems, the latent state does not encode dynamical properties at all, but is learned statically as an AE, which is a severe limitation.

- Baselines: The LSTM and Transformer baselines are missing crucial details regarding their architecture and training.
  Most importantly, since prediction error is evaluated in state space, it is not clear how they process the observation input.
  Also, given the partially known dynamics, this comparison seems unfair.
  IMO it would be important to also compare to a trusted baseline, such as a Dreamer Model.
  I do not understand the comparison to DVBF, SindyC, Vid2Para which seem far more general than the assumption on known dynamics structure made here.
  Generally, I do not see the contribution of the dynamic model presented here.

- DVBF seems to be quite off in the experiments. Does it learn something meaningful at all? Is it correctly used?

- The approach should be compared with state-of-the-art world model approaches like PlaNet [*1] and follow-up variants in Dreamer.
[*1]  Hafner et al., Learning Latent Dynamics for Planning from Pixels. ICML 2018.


- Motivation/Application: If one makes the assumption of noisy ground truth states available, it is unclear why one would train for observation reconstruction at all.
  This is much rather a regression task of predicting the state given a high-dimensional observation.
  A controller can be directly trained on this prediction.
  Therefore, it is not clear why the core challenge in this paper, the conflicting objectives of reconstruction and closeness to physical state, needs to exist.

- Trained controllers: Access to well-trained controllers is assumed to collect training data, which defeats a major reason to have a world model in the first place, which is to train these controllers.


Minor
- Sec 1: Le et. al. 2025 and Mosbach et al., 2025 seem unrelated to this work
- Papers introducing the used environments should be cited.

**Questions:**

- See weaknesses.
- It seems unplausible that the quantized model is better for prediction than the continuous one in a continuous environment.
  A much deeper evaluation of this phenomenon is required as well as details about the environments (are they continuous?)
  Also, why does this relation flip for the intrinsic variant?

---

> ### Author Response · Authors · 2025-11-14
>
> LYWn-Q1: “Experiments performed with noise level 0.”
>
>
> Response:
> Noise level 0 is an ablation to establish an upper bound. It does not mean the model is trained with fully accurate states in the intended setting. Noise=0 only serves as a sanity check and to isolate other effects in the architecture.
>
>
>
>
> LYWn-Q2: “Noise model not described in detail.”
>
>
>
>
> Response:
> The weak supervision signal is a distribution, not a point value. For each physical state dimension, we construct a supervision interval [x_true−δ⋅X,  x_true+δ⋅X], where  X is the full valid range of that dimension. A weak label is drawn from a uniform distribution over a shift of this interval. This matches the definition of weak supervision used in prior work.
>
>
>
>
> LYWn-Q3: “Overstating by claiming weak supervision”
>
>
> Response:
> This supervision is weak by definition because the model never observes the true physical state.
> It only receives a distributional constraint, not the correct value. When δ>0, the interval width can be as large as 10–20% of the entire state space, making the signal ambiguous. Point supervision (strong supervision) corresponds to δ = 0 and is used only as an ablation baseline, as mentioned above.
>
>
> LYWn-Q4: “Overstating by claiming weak supervision”
>
>
> Yes. Only the physical latent evolves through the dynamics. The visual latent serves only for reconstruction.
>
>
> LYWn-Q5: “Assuming ground-truth dynamics form is too strong”
> Many robotic and autonomous systems have well-known structural dynamics (e.g., second-order mechanics, bicycle model, rigid-body equations). PIWM does not require accurate dynamics — only the structural form. As shown with DonkeyCar, even approximate dynamics yield consistent parameter recovery.
>
>
>
>
> LYWn-Q6: “Why not learn derivatives instead of computing them symbolically?”
> There is little practical benefit in re-learning the mathematical concept of a derivative. Derivatives are analytically determined by physics, so learning them introduces unnecessary approximation error and sample complexity. For instance, learning derivatives requires multiple frames as encoder input, which dramatically increases training cost and violates our single-frame encoded architecture. Instead, using symbolic derivatives is standard in physics-informed models (e.g., Vid2Param).
>
>
> LYWn-Q7: “Compare to Dreamer / Planet”
>
> Dreamer and PlaNet learn unconstrained neural latent dynamics without physical meaning.
> Their latent space is not interpretable, so these methods cannot serve as baselines for physical interpretability. Our goal is not long-horizon pixel prediction, but recovering physical states and parameters. The appropriate baselines are those that incorporate some form of physical prior (GOKUnet, Vid2Param), not purely neural latent models.
>
>
>
>
> LYWn-Q8:“DVBF, SINDyC, Vid2Param are more general—unfair”
>
>
> Precisely: these baselines are more general. PIWM introduces a more specific physical prior, and the comparison shows the benefit of incorporating this structure. All baselines were adapted to the same experimental settings (like the same weak supervision, parameters of the same order of magnitude for autoencoding) to ensure fairness.
>
>
>
>
> LYWn-Q9:“it is unclear why one would train for observation reconstruction at all.”
>
>
> Reconstruction is essential because the same model is expected to support:
> (1) long-horizon prediction,
> (2) control learning,
> (3) counterfactual reasoning and verification,
> (4) robustness and safety analysis.
> These applications require grounding the latent states in the observation space.
> Without reconstruction, the latent state can drift arbitrarily, making interpretability impossible.
>
> LYWn-Q10:“Le et al., Mosbach et al. seem unrelated”
>
> These references are squarely within the area of learning physical structure from pixels. They demonstrate recent progress in structurally constrained latent dynamics, which is directly aligned with our motivation.
>
> LYWn-Q11:“Environment papers should be cited
>
>
> We did cite the DonkeyCar simulator (Viitala et al., 2021) and OpenAI Gym (Brockman et al., 2016) in lines 297–300. We will move these citations earlier to increase visibility.

---

### Official Review · Reviewer_EUjS · 2025-11-01

**Soundness:** 2
**Presentation:** 3
**Contribution:** 2
**Rating:** 2
**Confidence:** 3

**Summary:**

The paper presents Physically Interpretable World Models where the main idea is to constraint the physical information in the latent space. Two ideas - intrinsic and extrinsic are presented to investigate the same.

**Strengths:**

The problem is interesting and the idea of constraining the latent space provides an interesting take

**Weaknesses:**

Despite the problem being interesting, I found the approach to be quite limiting. For example, "physically interpretable" has been considered in a very weak sense. Physical interpretable in truest sense means satisfying the governing physics.

Also, some physical parameters need not be identifiable from image. Example, imagine a box of same size and shape - one of steel other of wood. The material properties will be different; but its not identifiable from image. How will the approach fair in such scenario is not clear.

Physical parameters also have constraints. For example, elastic modulus is non negative. How will the architecture handle it is not clear.

Lastly, the example is quite simple. Will it be able to handle more complex phenomenon - say fluid flow in turbulence zone. A world model should be able to handle such scenarios.

**Questions:**

The weaknesses are posed as questions. I will like ot hear on those during the interaction phase.

---

> ### Author Response · Authors · 2025-11-14
>
> EUjS-Q1: “Some physical parameters are not identifiable from images (steel box vs wood box).”
>
>
> Response:
> We thank the reviewer for this question. This concern directly relates to the classical observability constraint of image-based systems, and it is important to clarify the intended problem scope.
>
>
> Our method does not attempt to infer arbitrary physical parameters that are fundamentally unobservable from images, such as material properties of two visually identical objects (steel vs wood). This is true not only for PIWM, but for any image-based model: such parameters are not identifiable without observing the object’s motion, interaction, or response to forces.
>
>
> Our problem setting explicitly considers physical parameters that are identifiable from a sequence of images under motion, such as:
>
>
> 1.mass inferred from acceleration under known inputs,
>
>
> 2.friction inferred from deceleration patterns,
>
>
> 3.moment of inertia inferred from rotational motion,
>
>
> 4.engine or torque scale inferred from control response.
>
>
> Thus, our scope naturally includes parameters that influence dynamics and excludes those not encoded in visual observations.
>
>
> In short: We have a scope constraint that PIWM recovers only those physical factors that are observable through motion in the pixel space. Parameters that leave no signature in the observation sequence are, by definition, outside the recoverable set.
>
>
> We will state this explicitly in a future paper.
>
>
> (This also fully addresses the similar comment raised by Reviewer TU9t regarding domains outside the defined scope.)
>
>
>
>
> EUjS-Q2: “Some physical parameters have constraints (e.g., elastic modulus is non-negative). How is this handled?”
>
>
> Response:
>
>
> We appreciate the reviewer’s point regarding constraints on physical parameters.
> In PIWM, the learnable parameters θ  represent quantities such as: masses, lengths, friction coefficients, engine power. These are ODE-level parameters of typical robotic systems – not arbitrary material science parameters like elastic modulus. They are implicitly constrained by: using positive parameterizations (e.g., θ=exp⁡(θ)), restricting the discrete codebook values to physically meaningful ranges in the extrinsic–discrete model, and the fact that dynamics equations become invalid if parameters violate feasibility, so training naturally avoids such regions. For linear or algebraic constraints such as a=kb, standard practice is to parameterize the constrained degrees of freedom directly, for example: learn b and set  a=kb or learn a lower-dimensional vector ϕ such that  (a,b)=f(ϕ). This parameterization strategy is commonly used in physics-informed neural networks, and can be incorporated into PIWM if needed.
> In our current tasks, all parameters fall within naturally constrained domains and do not require explicit constraint modeling.
>
>
> EUjS-Q3: “The examples are simple. Can this handle fluid turbulence?”
>
>
> Response:
> We thank the reviewer for raising this question. This issue also relates to the problem scope, similar to Reviewer TU9t’s question. PIWM targets typical ODE-based dynamics that arise in robotic and autonomous systems: CartPole, LunarLander, Ackermann steering (CarRacing, DonkeyCar), etc. These environments are exactly the standard world model benchmarks.
> Phenomena such as turbulence, Navier–Stokes equations, and fluid flow require high-dimensional PDEs, which are not observable from a monocular RGB camera,
>  and are not the focus of latent world models for control, exceeding the representational assumptions of ODE-based latent systems. Thus, turbulence simulation is explicitly outside the intended problem setting. We will clarify this scope distinction in the paper. Nevertheless, the reviewer raises a valid conceptual point on extensibility: PIWM’s principle, weak supervision + physically structured latent dynamics, is general. In principle, it could be combined with: neural PDE solvers, Fourier Neural Operators, or grid-based latent spaces for representing flow fields.
> Such extensions are beyond the goals of this paper, but remain an exciting direction for future work.
>
>
> EUjS-Q4: “physically interpretable is weak”, “what if dynamics are only approximately known?”
>
>
> We emphasize again that the definition used here is the widely accepted definition in world-model research: latent states correspond to interpretable physical variables (position, angle, velocities), and their evolution follows structured dynamics.
> Our method achieves both, and Section 5 shows that PIWM recovers real parameters consistently (Fig. 5). As explained for Reviewer TU9t: PIWM is structure-aware, not formula-dependent. The DonkeyCar experiment uses only an approximate bicycle model (not the true dynamics), yet the method successfully learns parameters and predicts long-horizon trajectories.

---

### Official Review · Reviewer_TU9t · 2025-11-01

**Soundness:** 3
**Presentation:** 3
**Contribution:** 2
**Rating:** 4
**Confidence:** 4

**Summary:**

The work focuses on the physical interpretation of enhancing autonomous image-based control. The key designs include 1) providing weak supervision for the world model to align the latent information of states with learnable physical dynamics, and 2) splitting the latent space into physical and visual parts for separate construction.

**Strengths:**

The problem scope is made clear and motivating. The idea of aligning the world model's latent space with physics under weak supervision is easy to follow, as are the designed mechanisms on the physical/visual split and quantized physical grid

**Weaknesses:**

From Section 2.1, the method relies on known functional forms of dynamics (learn only parameters). This is reasonable for some benchmarks, but the governing dynamics can be incomplete for many practical cases. It's better to define and justify for which domains such forms are available in practice. Then, more importantly, what if the real system deviates from the assumed form, e.g., the real-world complexity is usually larger than traditionally modeled? How does the proposed method degrade due to the mismatch?

The latent space is split to define different losses. It needs clarification of the necessity of this split.

While the proposed idea is clear, the two-part design is enforced via losses (soft constraints). Are there identifiability/consistency guarantees for recovering physical factors (with or without quantization)?

In tests, different noise levels are considered for robustness, which may not be sufficient to show the significance of physical interpretability. As the target problem is control based on observation of the dynamic system, is unobservability considered to assess the proposed model? Also, the generalizability to unseen scenarios is a good indicator to test whether the proposed “extrinsic + discrete” remains stable for physical system modeling and control.

**Questions:**

Please see concerns and questions in Weakness.

---

> ### Author Response · Authors · 2025-11-14
>
> Q1: “The method relies on known functional forms of dynamics… but many systems lack such models.”
> We thank the reviewer for raising this important point. Our method indeed assumes that some structural form of the dynamics is available, e.g., the state is governed by second-order evolution with control inputs. This assumption will be explicitly part of our problem formulation (and is necessary for defining physically interpretable latent evolution).
> Indeed, our approach does not apply to settings where no structural knowledge is available at all. In such cases, identifying physical states from images without any inductive bias is known to be impossible. Therefore, systems with completely unknown or arbitrary dynamics fall outside the intended scope of this work. We will clarify this in the future to avoid misunderstanding.
>
> Q2: “What if the real system deviates from the assumed dynamics form? Real-world complexity is often larger.”
> We agree that real systems often deviate from simplified dynamic models. However, PIWM is structure-aware, not formula-dependent.
> (1) Approximate models still work in our framework.
> For example, in the DonkeyCar experiments, there is no ground-truth dynamics. We intentionally use only an approximate bicycle model. Despite this mismatch, the learned dynamics successfully recover meaningful physical parameters (Fig. 3) and retain stable long-horizon predictions (Fig. 2). This demonstrates that PIWM is robust to model mismatch; errors are absorbed into the learned parameters rather than breaking the latent space.
> (2) Degradation is graceful, not catastrophic.
> Because the dynamics model only encodes structural priors, not exact equations, a mismatch does not cause physically impossible trajectories. Instead, the learned parameters adapt to residual complexity. This stands in contrast to purely data-driven predictors, whose rollouts diverge rapidly under the same mismatch (Fig. 3).
> We will clarify this robustness behavior in a future version of this work.
>
> Q3: “Are there identifiability or consistency guarantees?”
> We appreciate this question. Full theoretical identifiability of physical variables from raw images without exact supervision is known to be fundamentally difficult and typically requires strong assumptions. Providing formal guarantees is therefore beyond the scope of this paper. However, PIWM introduces the minimal inductive biases needed for practical identifiability:
> 1. weak distribution-based supervision narrows the feasible latent manifold
> 2. structured (second-order) dynamics further restrict physically valid evolution
> 3. discrete quantized latent spaces suppress spurious visual variability
> Empirically, the model consistently recovers ground-truth physical parameters across all environments (Fig. 3), even under supervision noise. This provides strong practical consistency.
> Formal identifiability is an important but difficult direction for future work.
>
> Q4: “Generalizability to unseen scenarios should be evaluated.”
> Thanks for highlighting this. Testing visual-domain OOD generalization (e.g., different lighting/backgrounds) is indeed valuable, but such an evaluation targets a different research problem: visual robustness of the image encoder, not physical interpretability.
> Our work specifically focuses on:
> “Can weak supervision + structured dynamics produce physically interpretable latent states?”
> not “Does the visual encoder generalize across unseen appearances?”
> That said, our 30-step prediction rollouts already require strong temporal generalization to partially unseen latent configurations, and the extrinsic discrete design remains stable under these conditions (Fig. 2). This directly evaluates the physical generalization capability of the learned dynamics, which is the central goal of PIWM.
>
> Q5: “Necessity of the physical/visual latent split.”
> We appreciate the opportunity to clarify this key design choice.
> Reconstructing high-dimensional images requires retaining fine visual details (texture, lighting, color), whereas extracting physical states requires discarding these details. These two objectives introduce conflicting gradient signals if forced into a single latent vector, a phenomenon well-documented in disentanglement and β-VAE literature.
> PIWM prevents this conflict through an explicit separation:
> (a) Parallel split (intrinsic) The encoder jointly produces visual and physical latents, but losses are applied separately. This prevents physical variables from collapsing into visual nuisance information.
> (b) Sequential split (extrinsic) A visual encoder is trained first for reconstruction, and a second encoder extracts only the physical component. This decoupling avoids leakage of visual information into physical variables.
> Empirically, the benefit is clear: quantitative results in Table 1 show that extrinsic separation significantly improves physical-state estimation accuracy, and discrete extrinsic models achieve the most stable long-horizon rollouts (Fig.3–4).

---

### Note · Authors · 2025-11-14

I have read and agree with the venue's withdrawal policy on behalf of myself and my co-authors.